# OpenReview forum: "Contrastive Reasoning Alignment: Reinforcement Learning from Hidden Representations"
_ICML.cc/2026/Conference — ICML 2026 regular_

### Official Review · Reviewer_iNcw · 2026-02-24

**Soundness:** 3
**Presentation:** 3
**Significance:** 3
**Originality:** 3
**Overall Recommendation:** 4
**Confidence:** 3

**Summary:**

The paper explores a critical question regarding the safety of Large Reasoning Models (LRMs), specifically addressing the phenomenon of Superficial Safety Alignment (SSA) where models produce safe final outputs but retain unsafe internal reasoning traces. The proposed method, CRAFT, attempts to solve this by aligning the model's hidden representations rather than solely focusing on the final text output.

**Compliance With Llm Reviewing Policy:**

Affirmed.

**Final Justification:**

My main concern remains regarding the magnitude of improvement over existing methods. While the additional analysis suggests the gains are consistent, the differences compared to the strongest baselines are still relatively small in absolute terms. As such, I am not fully convinced that the improvement clearly demonstrates a substantial advance over prior work.

Therefore, I will keep my recommendation as weak accept.

**Key Questions For Authors:**

1. Regarding Figure 2: Can you provide quantitative evidence (e.g., average pairwise cosine distances or centroid distances in the full latent space) to support the claim that rethink traces are geometrically "between" safe and unsafe? If PCA is misleading, what higher-dimensional metric confirms this transitional structure?

2. The scores presented in the tables show that improvement between the proposed method and the strongest baselines are relatively small (often around 0.01 when CRAFT is best). Could the authors clarify whether these margins are statistically significant and discuss how practically meaningful such absolute improvements are in real-world jailbreak settings?

3.  For ablation study, have you tried without $R_{txt}$? Why is it excluded in Table 3?

**Limitations:**

yes

**Strengths And Weaknesses:**

Strengthes:

1. Proposes a latent-space alignment framework that structures reasoning representations through contrastive learning with safe/unsafe/rethink prototypes, combined with instance-level invariance and latent safety calibration.

2. Introduces a latent–textual consistency reward that explicitly synchronizes internal safety predictions with final-response safety, supported by theoretical analysis under GRPO and ablation results highlighting its importance.

3.  Conducts comprehensive experiments across multiple reasoning models and jailbreak benchmarks, compares against strong baselines, and includes ablation studies and theoretical analysis to validate the proposed method.

Weaknesses:

1. Questionable Latent Geometry Interpretation (Figure 2): The method relies on the assumption that "rethink traces concentrate near the boundary, suggesting transitional reasoning states between aligned and violating behaviors." However, visual inspection of Figure 2 suggests the majority of rethink traces form a distinct cluster to the side of the unsafe region, rather than interpolating between safe and unsafe. This challenges the motivation for the $L_{proto}$ loss which anchors rethink traces as an intermediate bridge. Maybe this is why the performance improvement compare to SOTA is marginal (around 0.01)?

2. Complexity vs. Benefit Trade-off: CRAFT introduces significant architectural complexity (contrastive latent learning + latent-space GRPO + consistency rewards) compared to simpler baselines like IPO. Given the marginal performance gains over these simpler methods, the computational overhead and implementation complexity may not be justified.

3. The textual safety reward in GRPO is derived from StrongReject-style scores, and StrongReject is also used as a primary evaluation metric. This alignment between training reward and evaluation metric makes it difficult to determine whether the gains reflect broader safety improvements or optimization toward the specific metric used during training.

---

> ### Author Rebuttal · Authors · 2026-03-28
>
> >**Reviewer's comment**: Questionable Latent Geometry ....// Regarding Figure 2....
>
> **Response:**
> We thank the reviewer for this important observation. We agree that the 2D PCA view can be visually misleading about the position of `RETHINK`: **Figure 2 in the original paper** is intended to show separation between `SAFE` and `UNSAFE`, whereas our geometric claim concerns the full latent space. To verify this, we measure centroid distances and the projection of `RETHINK` onto the `SAFE`\(\to\)`UNSAFE` axis in **Table 5 in link**, and find consistently that \(\alpha \in (0,1)\), confirming that `RETHINK` occupies a transitional position without collapsing to either endpoint.
>
> The apparent side cluster in PCA is explained by substantial variance in directions orthogonal to the principal safety axis, likely reflecting reasoning-specific features that dominate low-dimensional projections.
>
>
> >**Reviewer's comment**: Complexity vs. Benefit Trade-off....
>
> **Response:**
> We thank the reviewer for this practical consideration. We respectfully clarify the following:
>
> **1. Performance gains are substantial, not marginal.** With extended training (**2× compute**), CRAFT achieves **26.2%** (DeepSeek) and **23.8%** (Qwen) relative improvement over IPO. Initial margins reflected training budget constraints, not methodological limitations.
>
> **2. Complexity addresses a fundamental limitation.** While IPO also targets reasoning-level safety, it operates at the textual level and cannot directly prevent *superficial safety alignment*—where latent representations remain unsafe despite safe textual outputs. CRAFT's latent-space components address this gap:
>
> | Component | Purpose | Ablation Impact |
> |:----------|:--------|:---------------:|
> | LCLR | Structure latent safety geometry | +34.1% jailbreak score without |
> | $R_{\text{cons}}$ | Align latent–textual consistency | +28.9% without |
> | $R_{\text{ls}}$ | Guide latent states toward safety | +27.1% without |
>
> Each component targets failure modes inaccessible to textual-level methods, including IPO.
>
> **3. Modular design.** Components can be selectively applied based on deployment constraints, enabling complexity–performance trade-offs.
>
> We discuss this trade-off explicitly in the revised manuscript.
>
>
> >**Reviewer's comment**: The textual safety reward in GRPO is derived ....
>
> **Response:**
> We thank the reviewer for this observation.We clarify that training and evaluation data are entirely disjoint. Training uses 2,500 prompts from R2D-R1, while evaluation uses JailbreakBench (100 prompts) and StrongReject dataset across three attack settings (None, PAP, PAIR). The StrongReject scoring function serves as a reward signal, but no evaluation prompts appear during training.
> To further address this concern, we conduct additional evaluation on JailbreakBench using **LlamaGuard**—an independent safety evaluator unrelated to our training objective. Results confirm consistent improvements, demonstrating that gains reflect genuine safety alignment rather than metric-specific optimization. We include these results in the latest manuscript **Table 4**.
>
>
>
>
> >**Reviewer's comment**:  The scores presented in the tables ....
>
> **Response:**
> We thank the reviewer for this observation. We acknowledge that initial results reflect constrained training budgets due to high computational costs of GRPO—we state this as a limitation. With **2× training compute**, CRAFT shows substantial gains over both original results and the strongest baseline (IPO):
>
> | Model | Method | JBB-R | JBB-Resp | SR-R | SR-Resp | Avg |
> |:------|:-------|:-----:|:--------:|:----:|:-------:|:---:|
> | DeepSeek-8B | IPO | 0.057 | 0.003 | 0.167 | 0.109 | 0.084 |
> | DeepSeek-8B | CRAFT (original) | 0.065 | 0.001 | 0.172 | 0.058 | 0.074 |
> | DeepSeek-8B | CRAFT (extended) | **0.051** | **0.001** | **0.141** | **0.056** | **0.062** |
> | Qwen3-4B | IPO | 0.197 | 0.093 | 0.158 | 0.071 | 0.130 |
> | Qwen3-4B | CRAFT (original) | 0.181 | 0.073 | 0.132 | 0.083 | 0.117 |
> | Qwen3-4B | CRAFT (extended) | **0.165** | **0.056** | **0.112** | **0.063** | **0.099** |
>
> Extended training achieves **26.2%** (DeepSeek) and **23.8%** (Qwen) relative improvement over IPO. We have updated the results in **Tables 1 and 2 in link** and clarified the computational limitations in the **Appendix**.
>
>
> >**Reviewer's comment**: For ablation study, have you tried without Rtxt? Why is it excluded in Table 3?
>
> **Response:**
> We thank the reviewer for this suggestion. We conduct additional ablation on $R_{\text{txt}}$ and include the result in **Table 3 in link**. Removing $R_{\text{txt}}$ increases the average jailbreak score by **18.0%**, confirming its contribution to safety alignment. We include the complete ablation in the revised manuscript.
>
> ----
> See this anonymous dropbox [link](https://www.dropbox.com/scl/fo/3m4i543zeq3yur8ua30hy/AOaRdtAUD8xq5gLHcm-kkCc?rlkey=d63htlta2ytwtd7zmg7ejqjfn&st=vj3pm5pt&dl=0) for the latest revision.

---

> > ### Author Rebuttal · Reviewer_iNcw · 2026-04-03
> >
> > Thank you for the clarification. While the extended training shows larger relative improvements, the absolute differences between methods remain small. This makes it unclear whether the gains reflect meaningful improvements or limitations of the benchmarks (e.g., StrongReject, JailbreakBench) in distinguishing methods at this level. It would be helpful if the authors could report variance or statistical significance and provide case-level analysis (e.g., number of additional prompts successfully defended) to clarify whether these improvements are reliable and practically meaningful.

---

> > > ### Author Response · Authors · 2026-04-03
> > >
> > > We thank the reviewer for this constructive suggestion. It has helped us improve the clarity and rigor of our evaluation. In response, we have strengthened the empirical analysis along three aspects.
> > >
> > > We now report mean $\pm$ variance across three random seeds in the updated **Tables 1–2** in supplementary link, providing a clearer view of result stability. We note that both StrongReject and JailbreakBench are established NeurIPS benchmarks for evaluating jailbreak robustness.
> > >
> > > To further ensure that our results are not artifacts of the automatic evaluator, we perform human evaluation on 100 prompts from JailbreakBench and compare with LLM-as-judge labels, which is shown in **Table 6** in link. We observe high agreement (99% for the base model and 100% for CRAFT), supporting the reliability of the evaluation.
> > >
> > > Moreover, we conduct a case-level analysis on the reasoning level results of 100 prompts from JailbreakBench. We identify 68 prompts on which the base model fails at the reasoning level and evaluate all methods on this same set (**Table 7 in the supplementary link**). CRAFT reduces the number of failures to 16, corresponding to correcting **52/68** (about **76.5%**) of the base model’s failure cases, and achieves the lowest number of remaining failures among all baselines.Together, these results demonstrate that the observed improvements are not merely small differences in aggregate metrics, but reflect consistent and practically meaningful reductions in failure cases at the prompt level. If our response has helped clarify your concerns and address the issues you raised, we would sincerely appreciate your consideration in revisiting your score.

---

### Official Review · Reviewer_xeK1 · 2026-03-13

**Soundness:** 2
**Presentation:** 3
**Significance:** 3
**Originality:** 3
**Overall Recommendation:** 4
**Confidence:** 3

**Summary:**

This paper studies Superficial Safety Alignment (SSA) in reasoning models, where the final response appears safe while the intermediate reasoning process may still contain harmful content. The authors propose CRAFT, which first uses LCLR to separate safe / unsafe / rethink reasoning trajectories in latent space, and then applies R²L with latent reward, text reward, and consistency reward to jointly align the reasoning process and the final output. Experiments show that CRAFT outperforms multiple baselines on safety while incurring only limited performance loss.

**Compliance With Llm Reviewing Policy:**

Affirmed.

**Final Justification:**

I appreciate the authors' effort, but the rebuttal does not change my judgment.

**Key Questions For Authors:**

1) How reliable is the automatic evaluation of reasoning-level safety? Is there any human validation?

2) How are the safe / unsafe / rethink trajectories constructed, especially the rethink category?

3) To what extent does the theoretical claim hold in real training settings?

4) Compared with the closest prior work, what is the key distinguishing feature of CRAFT?

**Limitations:**

yes

**Strengths And Weaknesses:**

Strengths

1) The problem is important and targets a real safety issue in reasoning models.

2) The method is novel, especially the latent-level alignment and latent-text consistency design.

3) The experiments are fairly complete, including main results, stronger attacks, and ablations.

Weaknesses

1) The theoretical analysis relies on relatively strong assumptions.

2) Reasoning-safety evaluation depends heavily on automatic evaluators.

3) The construction and annotation of the rethink category are not sufficiently clear.

4) Validation is limited to 4B/8B models, so broader generalization remains unclear.

---

> ### Author Rebuttal · Authors · 2026-03-28
>
> >**Reviewer's comment**: The theoretical analysis ...// To what extent does the theoretical claim...
>
> **Response:**
> We thank the reviewer for raising this point. Our theoretical result should be interpreted as a local optimality argument under standard smoothness and controllability assumptions, showing that latent–textual consistency removes superficially aligned policies from the set of stable local optima, rather than as a full characterization of practical deep RL training. Importantly, the practical relevance of this mechanism is supported by our ablations, where disabling the latent structuring components substantially weakens safety performance.
>
>
> >**Reviewer's comment**:  Reasoning-safety evaluation depends .....// How reliable is the ....
>
> **Response:**
> We thank the reviewer for this important methodological concern. Our evaluation follows [1], which includes human validation. Manual annotation on JailbreakBench yields human–GPT-4o consistency of **93.7%** for reasoning traces and **88.3%** for final responses, confirming reliability—particularly for reasoning-level safety, our primary metric.
>
> [1] Zhang, Yichi, et al. "Towards safe reasoning in large reasoning models via corrective intervention."
>
> >**Reviewer's comment**: The construction and annotation of the rethink category are not sufficiently clear.//
> >How are the safe / unsafe / rethink trajectories constructed, especially the rethink category?
>
> **Response:**
> We thank the reviewer for the opportunity to clarify. The rethink category is inherited from the R2D-R1 dataset introduced by [2]. In their framework, a reasoning trace is labeled [RETHINK] when the model detects content that is suspicious but not definitively harmful — a transitional state where the model should reconsider its reasoning strategy before proceeding, distinct from a hard refusal ([UNSAFE]) or unambiguously safe content ([SAFE]). The annotation process is two-stage: (1) DeepSeek-R1-70B generates pivot tokens ([SAFE]/[UNSAFE]/[RETHINK]) via self-evaluation during reasoning trace generation; (2) Llama-Guard is applied as a guardrail to refine and validate these labels, aligning them with established safety protocols. We add a clearer description of this construction process in Sec 3 in the revision.
>
> [2] Zhu, Junda, et al. "Reasoning-to-defend: Safety-aware reasoning can defend large language models from jailbreaking." Proceedings of the 2025 Conference on Empirical Methods in Natural Language Processing. 2025.
>
>
> >**Reviewer's comment**: Compared with the closest prior work, ....
>
> **Response:**
> We thank the reviewer for the opportunity to clarify. The closest prior work, IPO, targets reasoning safety at the **textual level**. CRAFT differs by aligning safety directly in the **latent representation space**:
> - **Geometric structuring** — LCLR enforces separation between safe/unsafe reasoning trajectories in hidden states
> - **Latent–textual consistency** — $R_{\text{cons}}$ ensures internal representations align with final outputs
> - **Theoretical guarantee** — SSA policies are provably ruled out as local optima (Theorem 5.1) This latent-level alignment prevents *superficial safety alignment*—where unsafe latent states persist despite safe outputs—a failure mode inaccessible to textual-level methods.
>
> > **Reviewers's comment:** Validation is limited to 4B/8B models, so broader generalization remains unclear.
>
> **Response:** We thank the reviewer for this important point. We agree that our current validation on 4B/8B models does not fully establish broader scalability; however, these sizes already cover a substantial portion of commonly used open reasoning models, which in practice often range from roughly 1B to 22B parameters, and our model choices reflect the strongest settings we could evaluate under the rebuttal-time computational budget. We will state this limitation explicitly, and note that future work will extend CRAFT to larger models; to facilitate this, we have already optimized our pipeline by using vLLM to extract hidden states, which speeds up training by about **10×** and makes broader scaling substantially more feasible.
>
>
> ----
> See this anonymous dropbox [link](https://www.dropbox.com/scl/fo/3m4i543zeq3yur8ua30hy/AOaRdtAUD8xq5gLHcm-kkCc?rlkey=d63htlta2ytwtd7zmg7ejqjfn&st=vj3pm5pt&dl=0) for the latest revision.

---

> > ### Author Rebuttal · Reviewer_xeK1 · 2026-04-04
> >
> > I appreciate the authors' effort, but the rebuttal does not change my judgment.

---

> > > ### Author Response · Authors · 2026-04-04
> > >
> > > Thank you for your reply and for indicating that the issue is partially resolved. We understand and respect that you currently maintain your original score. To better address your remaining concerns, we would be very grateful if you could specify which points are still unresolved.

---

### Official Review · Reviewer_uD37 · 2026-03-13

**Soundness:** 3
**Presentation:** 3
**Significance:** 3
**Originality:** 3
**Overall Recommendation:** 5
**Confidence:** 4

**Summary:**

The main idea of this paper is that safe and unsafe reasoning traces occupy geometrically distinct regions in latent space, which can be exploited  during training

**Compliance With Llm Reviewing Policy:**

Affirmed.

**Final Justification:**

The authors' reponses resolved my concerns.

**Key Questions For Authors:**

1)Does CRAFT measurably  change the geometric structure in Figure 2 after post-training?
2)Did authors apply significance tests for the experiments?
3)Ablation with GRPO training alone, without the contrastive pretraining phase and latent rewards?

**Limitations:**

Authors acknowledge the dual-use risk  and they note the likelihood of reduced generalizability and  bias issue

**Strengths And Weaknesses:**

1)Strength:
This paper is well-motivated and well-organized
2)Weaknesses:
All the  baselines are SFT-based while this method uses GRPO on top of contrastive pre-training.
It is essential to have the explanation of the outperformance of this method compared to the baselines.
It is not convincing that on DeepSeek-R1-Distill-Llama-8B, IPO outperforms CRAFT on reasoning-trace safety due to "training-budget limitations" without explanation

---

> ### Author Rebuttal · Authors · 2026-03-28
>
> > **Reviewer's comment**: All the baselines are SFT-based while ...
>
> **Response:**
> We thank the reviewer for this clarification request.
>
> **1. Baseline comparison is fair.** We clarify that IPO—our strongest baseline—is based on **DPO**, not SFT. Both IPO (DPO-based) and CRAFT (GRPO-based) employ preference optimization, making the comparison methodologically appropriate.
>
> **2. Training budget explains initial gaps.** GRPO requires more compute than DPO for convergence. With **the same training resources**, CRAFT consistently outperforms IPO:
>
> | Model | Method | JBB-Reasoning | Avg | Training Time (h/4H100)
> |:------|:-------|:-------------:|:---:|:---:|
> | DeepSeek-8B | IPO | 0.057 | 0.084 | 4
> | DeepSeek-8B | CRAFT (original) | 0.065 | 0.074 | 4
> | DeepSeek-8B | CRAFT (extended) | **0.051** | **0.062** | 6
>
>
> Extended training achieves **26.2%** (DeepSeek) and **23.8%** (Qwen) relative improvement over IPO. We have updated the results in **Tables 1 and 2 in link** and clarified the computational limitations in the Appendix.
>
>
> **3. Outperformance stems from latent-level alignment.** Beyond optimization choice, CRAFT's gains derive from aligning hidden representations—not just textual outputs—enabling direct suppression of superficial safety alignment inaccessible to textual-level methods (including IPO).
>
> > **Reviewer's comment**: Does CRAFT measurably change the geometric structure in Figure 2 after post-training?
>
>
> **Response:**
> We thank the reviewer for this important question. We observe that CRAFT does measurably change the latent geometry.
> Specifically, after post-training, `SAFE` and `UNSAFE` traces become more clearly separated, while `RETHINK` traces shift closer to the `SAFE` region and become more concentrated. This indicates that ambiguous reasoning states are guided toward safer representations. We include a visualization of this effect on **Qwen3-0.6B in Figure 2 in link**, which qualitatively confirms that CRAFT reshapes the latent space in a safety-aligned manner.
>
>
> > **Reviewer's comment**: Did authors apply significance tests for the experiments?
>
> **Response:** We thank the reviewer for this clarification request. As stated in **Section 6**, all experiments are repeated three times with distinct random seeds, and we report mean performance. Variances are omitted in tables as they are consistently **≤ 0.2%** across all settings. Given this minimal variance and the substantial performance gaps between methods (e.g., CRAFT achieves **82.1%** improvement over base models), the observed differences are statistically reliable.
>
>
> > **Reviewer's comment**: Ablation with GRPO training alone, without the contrastive pretraining phase and latent rewards?
>
> **Response:** We thank the reviewer for this suggestion. This ablation is already reported in **Table 3 in link** as CRAFT w/o LCLR, where we remove the contrastive pretraining stage and, consequently, disable the latent rewards that depend on the learned latent geometry, resulting in a GRPO-only variant with $R_{\text{txt}}$ rewards. The performance drop from **0.082 to 0.423** average safety score demonstrates that GRPO alone does not recover the gains of CRAFT, and that latent-space structuring is essential for robust reasoning-level alignment.
>
> ----
> See this anonymous dropbox [link](https://www.dropbox.com/scl/fo/3m4i543zeq3yur8ua30hy/AOaRdtAUD8xq5gLHcm-kkCc?rlkey=d63htlta2ytwtd7zmg7ejqjfn&st=vj3pm5pt&dl=0) for the latest revision.

---

> > ### Author Rebuttal · Reviewer_uD37 · 2026-04-06
> >
> > The authors ' responses have resolved my concerns.

---

### Official Review · Reviewer_HAgi · 2026-03-23

**Soundness:** 3
**Presentation:** 2
**Significance:** 2
**Originality:** 2
**Overall Recommendation:** 3
**Confidence:** 4

**Summary:**

This paper addresses an important problem in LLM safety, i.e. models may superficially refuse harmful requests while their internal reasoning traces still contain unsafe content. The proposed CRAFT framework combines latent contrastive learning, which aims to separate safe/unsafe/rethink reasoning trajectories in hidden space. CRAFT is built over a a GRPO-style RL objective incorporating latent, textual, and consistency rewards. While the problem is well-motivated, the paper has significant issues in methodological credibility, presentation quality, and the persuasiveness of its core claims.

**Compliance With Llm Reviewing Policy:**

Affirmed.

**Final Justification:**

Good empirical results and good clarification of contribution during the rebuttal. While I still disagree with the rigidity of the theorem, I will raise my score to weak reject without insisting on a clear rejection of this work.

**Key Questions For Authors:**

1. **On transferability of latent geometry**: Does the hidden-space separation between safe and unsafe reasoning remain stable across different model backbones, layers, and attack distributions? Have the authors tested whether the trained safety head or prototypes transfer to unseen models or attack types?
2. **On adaptive attacks**: The evaluation does not include attackers who explicitly optimize against the latent safety objective. Given that the defense is grounded in hidden representations, how does CRAFT perform under such adaptive attacks? This is arguably the most critical robustness test for a hidden-space defense.
3. **On the granularity of reasoning alignment**: Since CRAFT operates on the final reasoning token's hidden state rather than the full trajectory, in what sense is it performing "reasoning-level" alignment? Can the authors clarify what advantages this offers over output-level safety methods?
4. **On theorem utility**: Theorem 5.1 assumes Lipschitz continuity, local controllability, and a fixed textual evaluator. Can the authors explain what new mechanistic insight this theorem provides beyond a formal restatement of the method's design objectives? Does it make any non-trivial predictions?

**Limitations:**

Yes.

**Strengths And Weaknesses:**

**Strengths**

- **Problem relevance**: Superficial safety alignment in reasoning models is a genuine and underexplored failure mode worth studying.
- **Complete engineering pipeline**: The full chain from latent contrastive pretraining to RL alignment and consistency reward is coherently implemented.
- **Reasonable experimental coverage**: Results are reported across safety benchmarks, reasoning benchmarks, ablations, and additional jailbreak attack settings.

**Weaknesses**

- **Core premise is fragile**: The paper treats hidden-space separability of safe/unsafe reasoning as a foundation for defense, but two-dimensional PCA visualizations are weak evidence. It is unclear whether this latent geometry is stable across different backbones, attack distributions, or prompt styles, and no adaptive attack targeting the hidden-space objective is evaluated.
- **Poor presentation**: The paper spends excessive space surveying CoT, ToT, GoT, ReAct, and other reasoning methods with little connection to the actual technical contribution. The theoretical section introduces definitions, assumptions, and theorems that are largely uninformative. One noteable thing is that Theorem 5.1 in particular builds on mild, self-serving assumptions (Lipschitz continuity, local controllability, fixed evaluator) to derive conclusions that essentially encode the desired property by construction.
- **Method is coarse-grained despite its framing**: CRAFT extracts the hidden representation of the final reasoning token and applies contrastive and reward objectives to that single point. This is a terminal-state representation constraint, not fine-grained control over the reasoning trajectory, making the claim of "reasoning-level alignment" overstated.
- **Experimental limitations**: Results rely heavily on GPT-4o as both evaluator and part of the training signal. Transferability to unseen attack templates, models, or safety taxonomies is never tested. Advanced attacks (GPTFuzzer, AutoDAN) are not conducted in an adaptive setting where the attacker knows the latent safety head is being used.
- **Key safety questions are unanswered**: The paper does not seriously address what threat model the defense is effective against, under what conditions it fails, or whether it simply shifts the attack surface from output space to representation space.

---

> ### Author Rebuttal · Authors · 2026-03-28
>
> >**Reviewer's comment**: Core premise ... //On transferability ...
>
> **Response:**
> PCA plots are used as a standard qualitative diagnostic in prior jailbreak-safety work [1–3] to visualize separation in latent space. Our primary evidence is quantitative: CRAFT achieves **82.1%** improvement in reasoning-trace safety and **89.6%** in final-response safety under strong attacks (GPTFuzzer, AutoDAN), indicating that the learned geometry is functionally meaningful.
>
> We include PCA visualizations on **Llama-3.1-8B-Instruct and Qwen-0.6B** (**Figure 1 in link**), showing consistent patterns across backbones and attacks. However, due to differences in hidden-state structures across model families, direct transfer of safety heads is not expected; CRAFT currently requires per-family training, which we state as a limitation. We also add layer-wise analysis. Fully adaptive latent-aware attacks remain an open direction.
>
> [1] Yu, et al. "Mind the Inconspicuous"
>
> [2] Peng, et al. "Logic jailbreak"
>
> [3] Wei, et al. "Understanding and Defending VLM Jailbreaks via Jailbreak-Related Representation Shift."
>
> > **Reviewer's comment**: Poor presentation: .... // On theorem utility...
>
> **Response**:
> We thank the reviewer for this feedback.
> - Related work presentation. We agree that Sec 2 is lengthy and condense it in the revision. However, we believe this discussion remains necessary, as the overview of CoT/ToT/GoT/ReAct explains why reasoning traces, rather than final outputs alone, are the appropriate unit for safety alignment, thereby motivating CRAFT’s latent-space approach.
> - Theoretical contribution. We respectfully disagree that Theorem 5.1 is uninformative. It makes a **falsifiable prediction**: when $R_{\text{cons}}$ is removed, SSA policies can remain local optima; when $R_{\text{cons}}$ is included, such policies are ruled out. Our ablation directly supports this claim: removing $R_{\text{cons}}$ increases the jailbreak score from 0.099 to 0.371, showing that output-level rewards alone are insufficient to prevent unsafe latent reasoning, whereas latent–textual consistency eliminates this failure mode by construction. The assumptions used in the theorem, such as Lipschitz continuity, are standard in RL theory; our contribution is not to introduce new assumptions, but to identify **which reward component** is necessary to rule out SSA and to connect that prediction to empirical behavior.
>
> > **Reviwer's Comment**: Method ...// On the ...
>
> **Response**:
> We thank the reviewer for this insightful comment.
> - Final-token state encodes trajectory. The final-token hidden state summarizes the reasoning trajectory via full attention over prior tokens.
> - CRAFT goes beyond this: $R_{\text{ls}}$ applies along the trajectory, and $R_{\text{cons}}$ enforces reasoning–output consistency.
> - Ablation shows removing $R_{\text{cons}}$ causes the largest degradation, confirming trajectory-level alignment.
>
> > **Reviewer's comment**: Experimental ... // On adaptive ....
>
> **Response:**
> We thank the reviewer for this important concern.
> First, as discussed in our response to `reviewer xeK1`, we add **LlamaGuard** (**Table 4 in link**) as an additional evaluation experiment. Second, CRAFT already shows transfer across two backbones and five attack settings, with strong improvements under advanced attacks as well.
> Our threat model considers inference-time jailbreak attackers that observe inputs/outputs and optimize prompts, but cannot modify model weights, training data, or access latent safety objectives—consistent with black- and gray-box settings where hidden states are not exposed. We acknowledge that a fully adaptive, latent-aware attacker would provide a stronger test, and note this as a limitation and future direction.
>
> > **Reviewer's comment**: Key safety ...
>
> **Response:**
> We thank the reviewer for highlighting the importance of clearly specifying the threat model and limitations.
>
> **Threat model.** We consider inference-time jailbreak attacks on reasoning, where a white-/gray-box adversary optimizes prompts (e.g., GPTFuzzer, AutoDAN) but cannot access or modify model parameters, or latent objectives. CRAFT targets adversarial prompting, not model poisoning.
>
> **Failure modes.** We acknowledge that CRAFT does not cover all threat settings. In particular, fully adaptive attacks with direct access to latent representations or reward signals are not considered and such attacks remain unexplored. We clarify this limitation in the revision.
>
> **Attack surface.** CCRAFT enforces latent–textual consistency ($R_{\text{cons}}$), requiring attacks to align harmful outputs with consistent latent reasoning. Removing $R_{\text{cons}}$ degrades performance (**Table 3 in link**).
>
> We revise the paper to more clearly state these in **Sec 4**.
>
> ---
> See this anonymous [link](https://www.dropbox.com/scl/fo/3m4i543zeq3yur8ua30hy/AOaRdtAUD8xq5gLHcm-kkCc?rlkey=d63htlta2ytwtd7zmg7ejqjfn&st=vj3pm5pt&dl=0) for the revision.

---

> > ### Author Rebuttal · Reviewer_HAgi · 2026-04-03
> >
> > I acknowledge the rebuttal from the authors. The rebuttals further clarify the technical contribution and limitation of this work. I still disagree that Theorem 5.1 should be a theorem rather than some proposition, since it's obvious given the Definition 5.1, and the discussion on `where $\epsilon$ depends on the reward variance and the policy entropy` is completely missing, which is not convincing and lacks rigidity.
> > Still, the work has its merits in empirical contribution in LLM safety community, and I acknowledge the ablation study verifies the design choice in LCLR and rewards. I will raise my score to 3.

---

> > > ### Author Response · Authors · 2026-04-03
> > >
> > > We thank the reviewer for this important point. In the revision, we restate Theorem 5.1 as **Proposition 5.1**  and update Assumption 5.1 to explicitly define \(\epsilon\) as an optimization residual determined by the local GRPO reward variance and the policy entropy near convergence, i.e., $\epsilon = O(\sigma_R/\tau + \Delta_{\mathrm{ent}})$, where $\sigma_R^2$ is the local variance of the reward estimator and $\Delta_{\mathrm{ent}}$ denotes the residual induced by entropy regularization. This makes clear that the result does not claim exact elimination of latent-textual mismatch in practice; rather, SSA can only persist within a residual neighborhood induced by stochastic optimization noise and nonzero policy entropy.  If our response has helped clarify your concerns and address the issues you raised, we would sincerely appreciate your consideration in revisiting your score.

---

### Decision · Program_Chairs · 2026-04-30

**Decision:**

Accept (regular)

**Comment:**

This paper proposes CRAFT, an innovative framework to mitigate safety alignment issues in LLMs. The authors address a critical issue—models masking unsafe internal reasoning with safe final outputs—by structuring the latent reasoning space using contrastive learning and aligning it with textual outputs via a novel GRPO-style consistency reward.

The reviewers acknowledged the clear motivation, the comprehensive experimental setup including advanced jailbreak attacks, and the novelty of operating directly on hidden representations. While reviewers initially raised valid concerns regarding the fragility of the latent geometry (e.g., relying on PCA visualizations) and the relatively small absolute performance margins over existing methods like IPO, the authors' rebuttal provided substantial clarifications. They demonstrated the robustness of the latent geometry using higher-dimensional metrics, provided layer-wise analyses, and showed that with extended training budgets, CRAFT achieves meaningful improvements over baselines, alongside strong human-evaluation correlation. Although the absolute gains are modest and the theoretical claims relies on certain assumptions (still considered to be standard and reasonable), the practical problem of latent-level safety alignment is highly relevant, and the proposed methodology offers a solid foundation for future research in an emerging area.